# Circularly Polarized Broadband Printed Antenna for Wireless Applications

**DOI:** 10.3390/s18124261

**Published:** 2018-12-04

**Authors:** Md. Samsuzzaman, Mohammad Tariqul Islam

**Affiliations:** 1Centre of Advanced Electronic and Communication Engineering, Universiti Kebangsaan Malaysia, Bangi 43600, Malaysia; 2Department of Computer and Communication Engineering, Patuakhali Science and Technology University, Dumki, Patuakhali 8602, Bangladesh

**Keywords:** broadband, circular polarization (CP), compact, planar, sickle-shaped, wireless applications

## Abstract

A simple, compact sickle-shaped printed antenna with a slotted ground plane is designed and developed for broadband circularly polarized (CP) radiation. The sickle-shaped radiator with a tapered feed line and circular slotted square ground plane are utilized to realize the wideband CP radiation feature. With optimized dimensions of 0.29λ × 0.29λ × 0.012λ at 2.22 GHz frequency for the realized antenna parameters, the measured results display that the antenna has a 10 dB impedance bandwidth of 7.70 GHz (126.85%; 2.22–9.92 GHz) and a 3 dB axial ratio (AR) bandwidth of 2.64 GHz (73.33%; 2.28–4.92 GHz). The measurement agrees well with simulation, which proves an excellent circularly polarized property. For verification, the mechanism of band improvement and circular polarization are presented, and the parametric study is carried out. Since, the proposed antenna is a simple design structure with broad impedance and AR bandwidth, which is a desirable feature as a candidate for various wireless communication systems. Because of the easy printed structure and scaling the dimension with broadband CP characteristics, the realized antenna does incorporate in a number of CP wireless communication applications.

## 1. Introduction

Circularly polarized (CP) microstrip antennas have been popularly used in various contemporary wireless communications systems, such as for satellite communication, WLAN (Wireless Local Area Network) and WiMAX (Worldwide Interoperability for Microwave Access), GPS (Global Positioning System), RFID (Radio-frequency identification) tag and RFID readers because of the versatile orientation of the transmitter and receiver [1,2]. Systems utilizing CP antennas in both transmitter and receiver ends have observed they improved upon cross polarization discrimination and present better immunity to multipath propagation. With the latest interest in high-speed transmission for wireless communications, the marketplace demand for CP antennas with a broad impedance and AR (Axial Ratio) bandwidth are in popular.

Several types of antennas with different shapes for the patch, feed line, and different slots in the ground and radiating patches, which can generate considerable wideband CP properties, have been reported in the literature. Among the reported research, dielectric resonator antennas (DRAs) [3,4,5], cross dipole antennas [6,7,8,9] and magneto-electric dipole antennas [10,11,12] was designed for broadband CP characteristics. The DRA designed by [3] to achieve ARBW (Axial Ratio Bandiwdth) of 46% while the magneto-electric dipole antenna proposed by [12] yielded IBW (Impedance Bandwidth) of 65%, 3 dB ARBW of 71.5%. However, the thickness of DRA is comparatively large and complex design structure, and a dipole antenna requires a distance of quarter wavelength from the radiator to the ground plane, which makes these antennas relatively bulky. To solve the issue, array antennas [13,14] and multilayer antennas [15,16] were suggested. However, array antennas require a complex feeding network while multilayer antennas pose complexity in the alignment and integration process. Besides, these type of antennas suffers narrow bandwidth. To minimize the above drawbacks, various recently reported CPW (Coplanar Waveguide) antennas, monopole antennas, reactive impedance surface based UWB (Ultra Wideband) antennas and different shape slot antennas were studied [16,17,18,19,20,21,22,23,24,25,26,27,28,29,30,31,32,33,34]. To enhance the performance of the small antenna especially gain and impedance bandwidth, several researchers were used reactive impedance surfaced metamaterial [17]. Using planar-patterned metamaterial concepts, broad bandwidth and high gain rectangular patch antenna were designed [18,19]. A coupled capacitive-inductive circuit is formed using the patterned patch and ground plane. The dimension of [18] is 28 mm × 32 mm, and the bandwidth is covering from 5.3–8.5 GHz with the gain above 4 dBi. Rectangular radiating patches, which are asymmetrically positioned concerning the microstrip line, attain an impedance bandwidth of 136% and AR bandwidth of 77% with large dimensions [20]. An approximately 60% large AR bandwidth was obtained through the inverted L and U-shaped monopole strip and a slotted ground plane [21]. The 61.96% CP bandwidth has been attained with an inverted asymmetric arm L-shaped microstrip and slotted ground plane, but the dimensions of the antenna were large [22]. An asymmetrical Y-shaped feeding line is used to produce the CP radiation for X band applications and a through via in series with an inductive strip is employed to achieve an impedance bandwidth of 31.4% [23]. A wideband CP printed triangular monopole antenna is also proposed in [24]. By asymmetrical excitation of a trilateral ground plane and a planar triangular wideband circular polarization is achieved, which is covering 1.42 to 2.7 GHz (62%). Though the antenna does not cover a compact dimension. The overall dimension of the antenna is 79.2 × 112 mm^2^. A simple and compact broadband CP antenna was presented with a slotted ground plane and semicircular slot. But the antenna achieves impedance bandwidth of 116.6% (2.37–9 GHz), and AR bandwidth of 58.7% (3.22–5.9 GHz) with 0.71.3λ × 0.71.3λ × 0.02λ at low frequency [25]. A moon shaped [26], a wide slot antenna [27], and a chifre-shaped antenna [28] with 3-dB AR bandwidths of 40%, 49%, and 41.6% were offered, respectively. An inverted-L grounded strip with two asymmetric T-shaped feed lines was stated to attain 60% impedance and AR bandwidth [29]. Furthermore, the two Vivaldi antenna components were vertically and horizontally placed to diminish the antenna size and also to understand broadband CP operation [30] which is a comparatively complex design. Recently, an antipodal Y strip with a square slot antenna [31], cross-shaped printed antenna with extended ground plane [32], parasitic strips with a partial ground plane [33], CPW-fed open loop [34] and compact wideband directional CP antenna [35] have been presented to realize wide impedance and AR bandwidth. Nevertheless, above the investigations of different printed antennas, none of these antennas AR bandwidth exceeds over 70%.

In this article, a compact broadband sickle-shaped printed CP antenna excited by a straightforward tapered microstrip feed line is offered. The achieved overlapping 3 dB AR bandwidth is 73.33%, which is much larger compared to the bandwidth accomplished previously with comparable dimensions and style techniques. The fabricated antenna is easy in design structure, broad impedance (−10 dB) and 3 dB AR bandwidth and does apply for integration in a variety of multiband wireless applications. The realized antenna was first designed, simulated and optimized in Ansoft 3D electromagnetic high-frequency simulator (HFSS) and then validated by fabrication and measurement. Finally, we are scaling the dimension to check the impedance and CP property of the realized antenna.

## 2. Antenna Design Structure

### 2.1. Antenna Schematic Layout

The geometric layout of the realized sickle-shaped antenna is displayed in Figure 1, which is designed and fabricated on the low-cost epoxy resin (FR4) substrate with the relative permittivity of 4.5 and a dielectric loss of 0.02. The physical dimensions of the anticipated antenna are 39 mm × 39 mm × 1.6 mm, which is 0.29λ × 0.29λ × 0.012λ for the lowest −10 dB frequency. The anticipated antenna is comprised of a semicircular slotted patch with a tapered microstrip feed line on the upper side of the substrate and a circular slotted square ground plane on the other side of the substrate. This semicircular patch radiator is fed by a 50 Ω tapered microstrip line to create a sickle-shaped with a width of *Wf*_1_ and a length of *Lf*_1_ for signal transmission. A circle with center *O* and radius *R1* is imprinted out from the rectangular ground plane, which is responsible for the created wideband impedance bandwidth with linear polarization (LP) in the desired operating band. On the other hand, the semicircle patch is created by subtracting the circle (*B*, *R3*) from the circle (*O*, *R2*), where *A* and *B* define the center of the two circles and R is the radius of the circle. These three points (*O*, *A*, *B*) make an angle <*AOB*, which is defined by α. By tuning the angle α = 96°, the maximum impedance and AR bandwidth are acquired. The semicircular slotted patch and feed line make the shape appear as a sickle. The comprehensive dimensions of the optimum antenna configuration as exposed in Figure 1a are *L* = 39 mm, *Lf*_1_ = 4.15 mm, *Lf*_2_ = 5.14 mm, *W_f_* = 2.24 mm, *R1* = 17 mm, *R2* = 11.2 mm, *R3* = 7.7 mm, *h* = 1.6 mm, α = 96°, and *L1* = 25.42 mm.

### 2.2. Antenna Design Evolution Procedure

To validate the development procedure, three layouts is depicted, as shown in Figure 2 where Ant1 has exhibited an LP in the two resonant modes and CP of one resonant mode with circular patch antenna that is fed by a tapered microstrip line. To improve the impedance bandwidth of Ant1, an *R1* circular slot is etched out (Ant2). According to the S_11_ plot shown in Figure 3a, the etched out *R1* slot results in an improvement in the −10 dB impedance bandwidth from 2.2 to 9.4 GHz, except for the middle frequency of 4–5 GHz. The Ant1 is designed at multiple resonating frequencies, which gives some small bandwidth in the specific point. To enhance the impedance bandwidth, a circular slot, *R1* in the ground plane has been added which overlaps the fundamental mode of the antenna and creates higher order mode. This higher order mode also helps to overlap the original bandwidth, and that is responsible for gradually increases the overall bandwidth of the antenna from 2.2 to 9.4 GHz. However, it can be seen that both Ant1 and Ant2 have an LP radiation wave (AR values greater than 9 dB) except at 6.2 GHz of Ant1. The additional step to stimulate the electrical field parts with identical amplitudes and phase variations of 90° to create the CP radiation. To achieve the CP features, a semicircular slot with a radius *R3* is etched out from the circular patch radiator, seen as Ant3 in Figure 2. It can be observed from Figure 2 and Figure 3 that *R3* in the patch radiator plays an important role to create wide 3 dB AR bandwidth with improved impedance bandwidth. These advancements are because of the growth in the lengths of the current path. The AR is significantly diminished from 20 dB for Ant1 to underneath 3 dB for Ant3. From Figure 3, it could be noticed that without the *R3* slot, the antenna achieves a 125% (2.20–4.24 GHz, 5.32–10.2 GHz) impedance bandwidth (−10 dB) and 0% dB AR bandwidth. On the other hand, with the R3 slot, the proposed designed attained a 122.80% (2.2–9.20 GHz) impedance (−10 dB) and 69.71% (2.28–4.72 GHz) 3 dB AR bandwidth.

### 2.3. CP Mechanism Analyze

To justify the CP property mechanism, the computer-generated surface electric current distribution of the anticipated semicircular radiating patch at 2.6 GHz in the Z > 0 direction is displayed in Figure 4, where Jsum characterizes the vector sum of all the significant current distributions. Jsum at t = 0 is orthogonal compared to that in t = T/4 and alternates clockwise as the period increases in the regular interval, thus generating the left-hand circular polarized (LHCP) radiation in the +Z direction.

### 2.4. Various Parameter Optimization

To investigate the consequences of the corresponding structural variables on the impedance and AR bandwidth, parametric research has been completed in this section. Figure 5 portrays the simulated S_11_ and AR with dissimilar parameter values of *R1*, *R2, R3, L1*, and angle α. Initial values were fixed of all the parameters during optimization.

(a) Effect of ground plane circular slot radius *R1*

Figure 5a depicts the antenna functionality against the bottom circular slot radius *R1*. As displayed in Figure 5a, *R1* includes a nominal influence on the impedance bandwidth of higher frequencies when its radius decreases. On the other hand, increasing its radius from an optimum value of 17 mm improves the S_11_ in the higher frequency range but upshifts, the lower frequencies. However, the variations in *R1* have a nominal influence upon CP procedure in the higher frequency range. Nevertheless, when *R1* is varying with regards to the 17 mm radius, the antenna seems to lose its CP property in the low and middle of the targeted frequency range.

(b) Effect of patch circular radius *R2*

The proposed antenna functionality is also examined when the patch circle radius *R2* is diversified as depicted in Figure 5b. The impedance bandwidth is relatively susceptible to the deviation of *R2*. From Figure 5b, it can be noticed that reducing *R2* with esteem to the ideal 11.2 mm causes the antenna to reduce its −10 dB bandwidth while raising it causes the antenna S_11_ to reduce in the upper rate of the recurrence range. However, the variants of *R2* have a nominal influence on the CP radiation in the larger frequency range. Nevertheless, when *R2* is reduced with regard to the most effective worth, the antenna seems to lose its CP procedure in the low operating frequency range.

(c) Effect of patch circular slot radius *R3*

Figure 5c describes the overall antenna performance against the circle slot patch radius *R3*. As demonstrated in Figure 5c, *R3* includes a less significant influence on the impedance bandwidth when its size decreases. Nevertheless, increasing its size decreases the S_11_ in the center frequency range. However, all the CP settings are sensitive to the patch slot radius *R3*. The circular slot radius *R3* can influence the relative placement of the sickled slot patch strips. Therefore, the CP behavior is usually influenced because of the coupling capacitance between the sickle from the patch and the copper in the bottom plane. Seen from the graph that reducing *R3* with regard to 7.7 mm triggers the antenna to drop its CP property completely within in the entire bandwidth while increasing it triggers the antenna to reduce its CP features in the low-frequency range. Seen from Figure 5c that the broadest 3 dB ARBW is definitely accomplished when *R3* is 7.7 mm.

(d) Effect of patch circular slot radius *L1*

Figure 5d displays the overall antenna performance versus the antenna feed placement size *L1*. The monopole feed line is the primary radiator of the antenna. Thus, the semicircle slotted patch mounted on it plays an essential role in the antenna linear radiation design for CP radiation features. As demonstrated in Figure 5d, *L1* has a nominal influence upon impedance bandwidth when *L1* decreases from its optimum value. However, increasing the space improved the S_11_ in the upper rate of the recurrence but decreases the low-frequency range. However, the variants of *L1* have minimal influence on the CP procedure in the desired working band.

(e) Effect of <AOB angle *α*

Figure 5e depicts the antenna performance versus the <AOB angle *α.* From this figure, the S_11_ is sensitive to the variation of the three-circle angle α. The S_11_ at the lower frequency bands tends to decrease as α increase, while very small changes are found at the higher frequencies. However, when the angle decreases from the optimum value, the lower frequency does not affect while the higher frequency S_11_ decreases. However, variations of α have a minor aftereffect of the AR based on the desired operating band.

## 3. Experimental Result and Discussion

To validate the feasibility of the proclaimed broadband CP antenna, a constructed prototype was measured and fabricated. The experimental results are measured with an Agilent N5227A PNA microwave network analyzer (Keysight, CA, USA) and a UKM Satimo StarLab (Microwave Vision Group, French). The snapshots of the fabricated antenna are verified in Figure 6a. From the realized results portrayed in Figure 6a, the measured impedance bandwidth of the reflection coefficient below the −10 dB is approximately 7.70 GHz (2.22–9.92 GHz), which is approximately 126.85% with regard to the center frequency at 6.07 GHz. On the other hand, the measured 3 dB AR bandwidth is around 2.64 GHz (2.28–4.92 GHz), which is approximately 73.33% at 3.60 GHz in Figure 6b. The measured and simulated radiation patterns at 2.40 GHz, 3.50 GHz and 4.30 GHz are plotted in Figure 7. The antenna radiates a bidirectional wave with opposite sense circular polarization. The radiation pattern of the antenna is usually LHCP (Left Hand Circular Polarization) for Z > 0 and RHCP ((Right Hand Circular Polarization) for Z < 0. It can be seen that the primary beam direction hardly shifts away from the zenith at both frequencies. The measured and simulated broad sight gains and antenna efficiency are shown in Figure 8. The average measured peak realized gain is greater than 3.98 dBic in the realized CP operating band and the maximum peak realized gain is around 4.85 dBic at 4.92 GHz. The antenna efficiency of the archived 3 dB AR frequency band range is approximately 80%.

Table 1 describes the overall performance of the realized antenna with existing literature reviewed antennas in the literature review section. Here **λ_0_** refers to the wavelength of the lowest frequency band for each reported antenna. It can be clearly noticed that the realized antenna achieves the widest fractional bandwidth in terms of dimension, impedance, and AR bandwidth property. Table 2 reveals the impedance and axial ratio bandwidth performance with applications of the recommended design for the various scaling dimension (L) just., It could be observed from the presented table that the wide impedance (a lot more than 75%) and AR bandwidth (more than 66%) have already been accomplished for the scaling dimension (L) without varying the additional antenna parameters. If the realized parameters, such as *R1*, *R2*, *R3*, *L1*, and *α*, could be tuned, the impedance and AR bandwidth may also be improved. Therefore, any researcher could be very easily remodeled from the recommended created for any preferred wide CP working band application through the scaling dimension (L) without varying the form, which may increase the sensor journal reader interest of the offered antenna.

## 4. Conclusions

In this article, a novel compact broadband CP sickle-shaped structure excited with a simply tapered feedline has been designed, fabricated and experimentally verified. The broad −10 dB impedance and 3 dB AR bandwidth performance is achieved due to the sickle-shaped patch radiator and the circular slotted square ground plane. The entire functionality for distinctly designed parameters has been analyzed and maximized. Through parameter optimization, the measured 3 dB AR bandwidth is 73.33% (2.28–4.92 GHz), and the impedance bandwidth is 126.85% (2.22–9.92 GHz), which agree well with the simulation results. The realized antenna is compact with design simplicity, broadband CP property, and low cost, making it suitable for the practical integration in WLAN/WiMAX and other broadband wireless communication systems. Finally, we have scaling the dimension to check the impedance and CP property of the realized antenna.

## Figures and Tables

**Figure 1 sensors-18-04261-f001:**
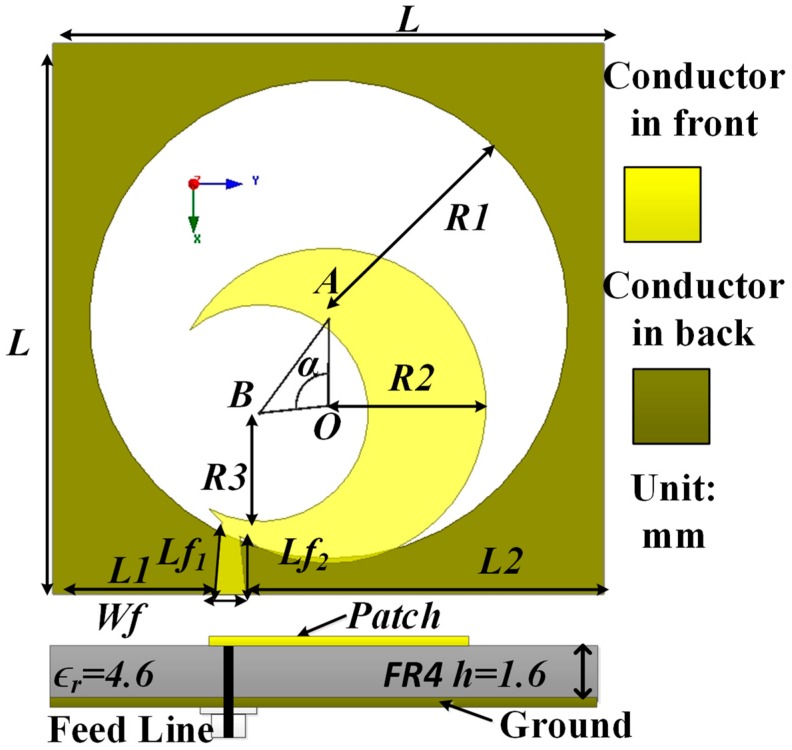
Geometric layout and cross-sectional view of the CP antenna.

**Figure 2 sensors-18-04261-f002:**
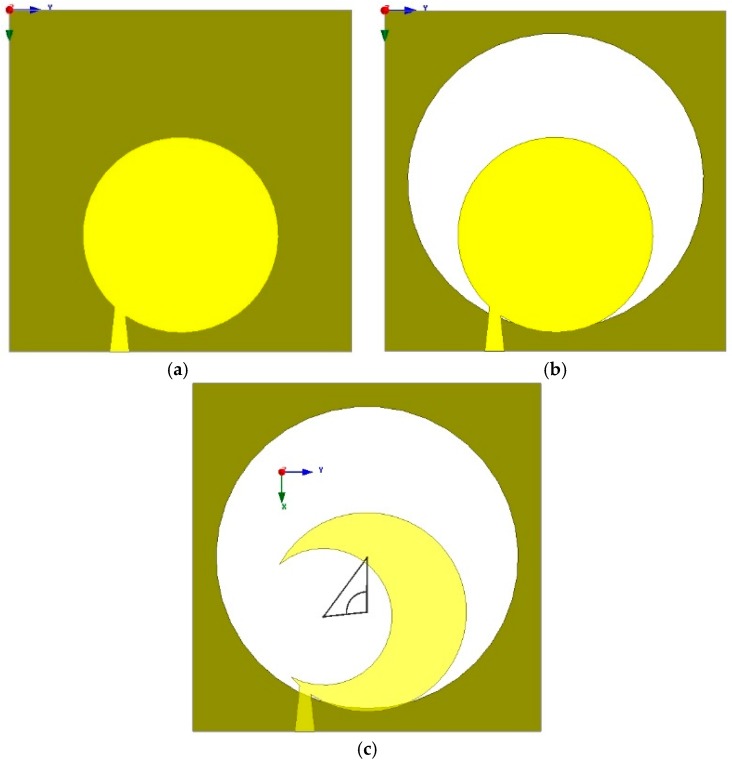
Evolution of design: (**a**) Ant1 (**b**) Ant2, (**c**) Ant3.

**Figure 3 sensors-18-04261-f003:**
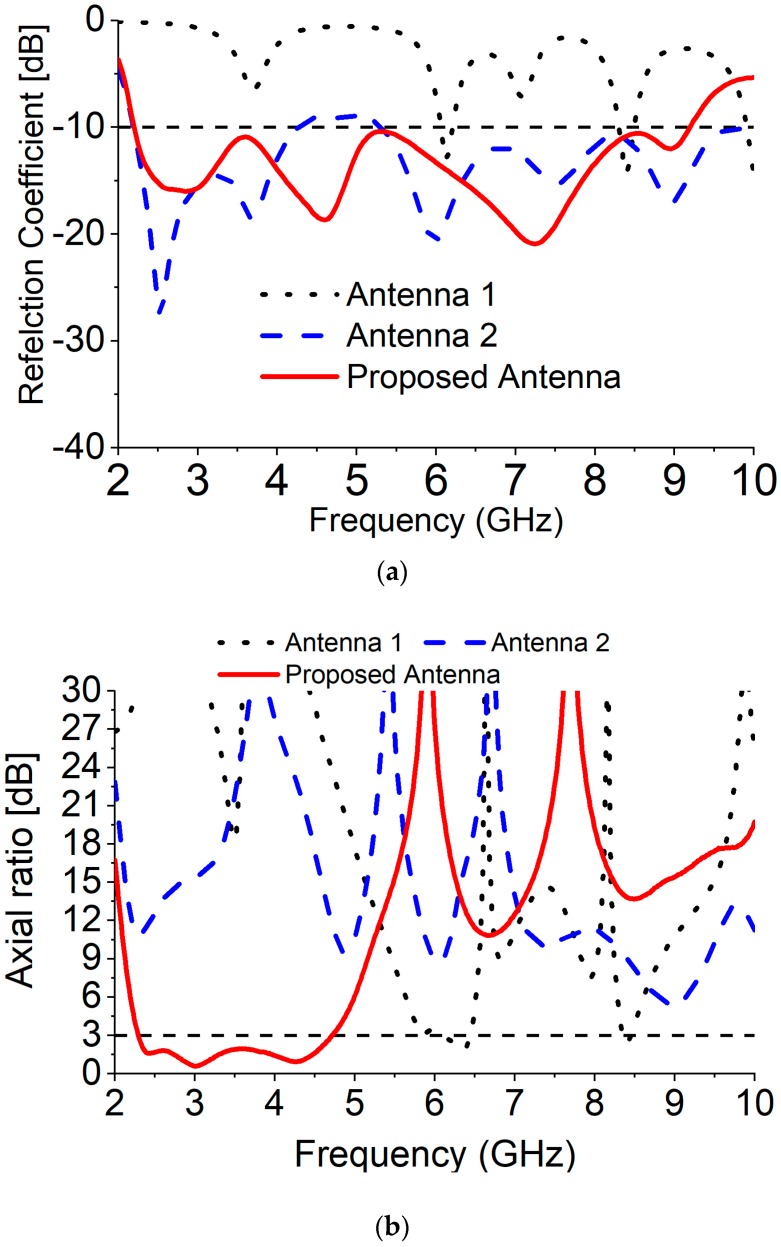
Effects of antenna evaluation: (**a**) reflection coefficient (S_11_), (**b**) AR.

**Figure 4 sensors-18-04261-f004:**
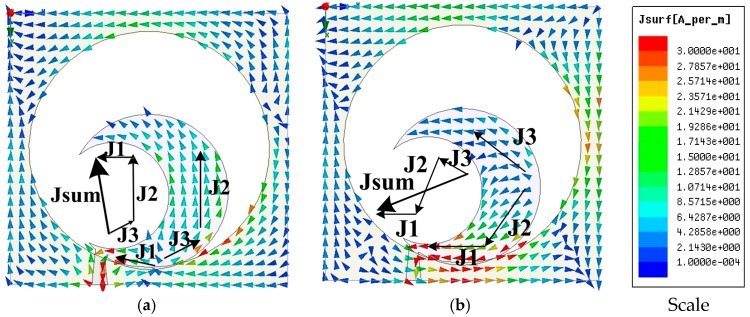
The simulated surface current distributions at 2.60 GHz (**a**) t = 0 (**b**) t = T/4 (**c**) t = T/2 (**d**) t = 3T/4.

**Figure 5 sensors-18-04261-f005:**
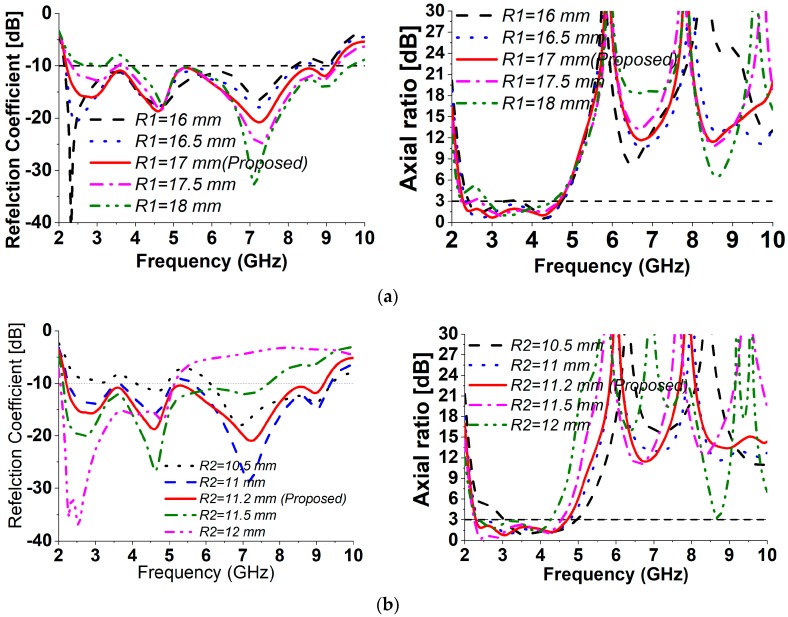
Numerical S_11_ and AR performance with different values of (**a**) *R1*, (**b**) *R2*, (**c**) *R3*, (**d**) *L1*, (**e**) α.

**Figure 6 sensors-18-04261-f006:**
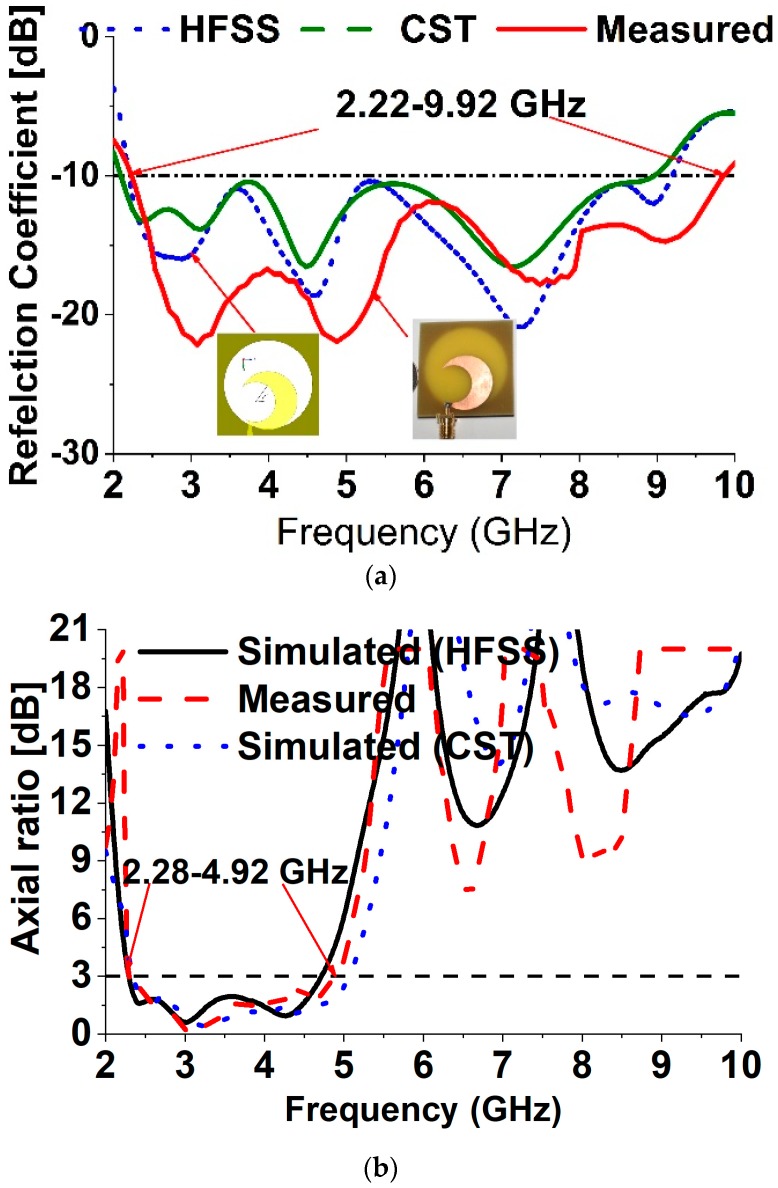
The numerical and measured results of (**a**) S_11_ (Reflection coefficient) and (**b**) ARBW.

**Figure 7 sensors-18-04261-f007:**
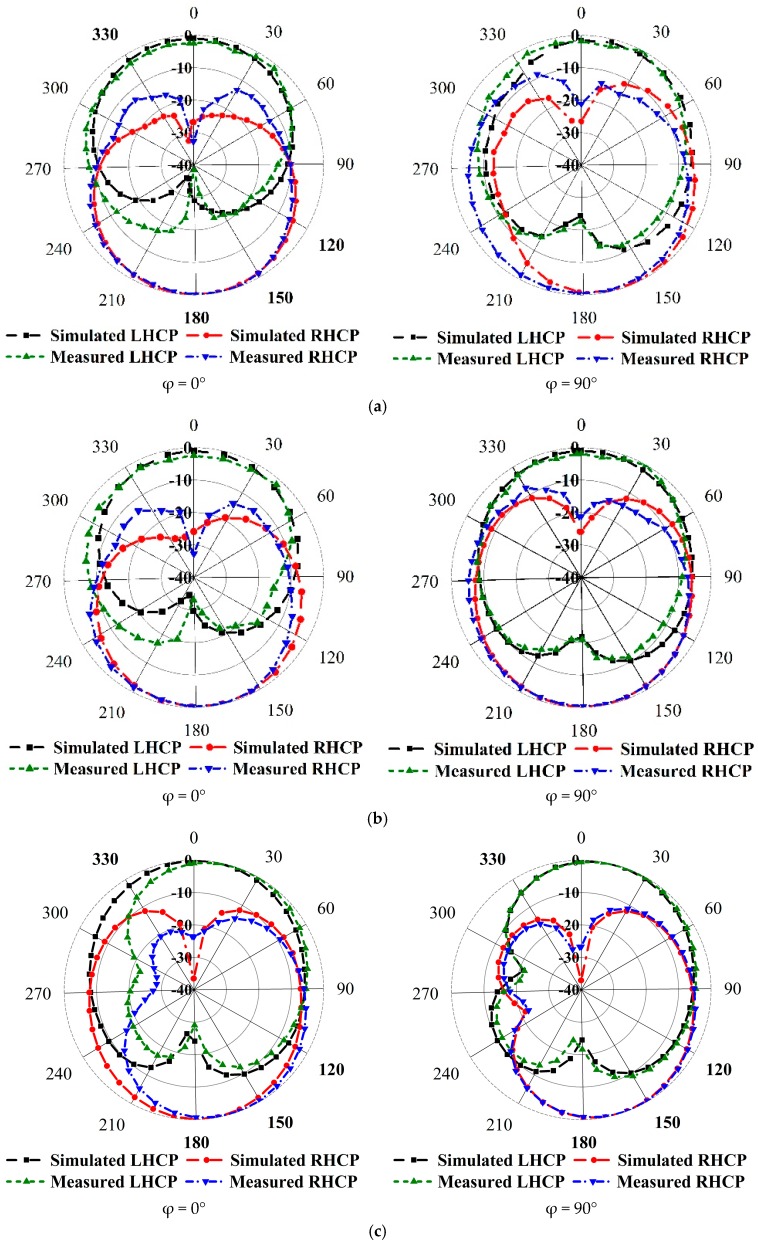
The numerical and measured CP radiation patterns at (**a**) 2.4 GHz (LHCP and RHCP) (**b**) 3.5 GHz (LHCP and RHCP) and (**c**) 4.3 GHz (LHCP and RHCP).

**Figure 8 sensors-18-04261-f008:**
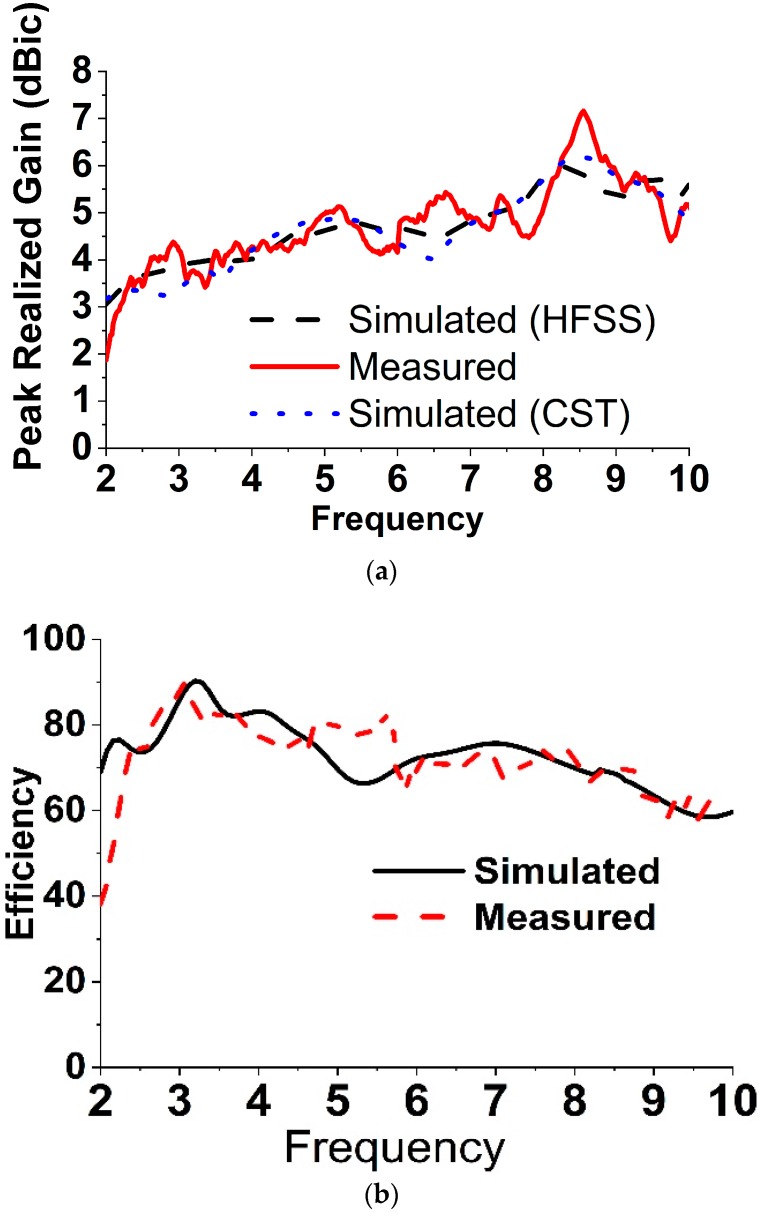
The numerical and measured (**a**) peak realized gain and (**b**) efficiency.

**Table 1 sensors-18-04261-t001:** Comparison different structures presented in ref.

Ref.	Year	f_c_ (GHz), f_cAR_ (GHz)	Electrical Dimension (λ_0_^3^)	Relatives BW	AR Relatives BW
[20]	2012	2.65, 2.45	0.63 × 1.45 × 0.013	136	77
[21]	2012	3.02, 2.9	0.48 × 0.44 × 0.015	67.76	62.07
[22]	2014	2.54, 2.69	0.494 × 0.494 × 0.014	55.40	61.96
[23]	2014	10.85, 10.85	0.5 × 0.5 × 0.018	31.4	31.4
[25]	2015	5.685, 5.685	0.71.3 × 0.71.3 × 0.02	116.6	58.7
[24]	2015	2.06, 1.97	0.61 × 0.54 × 0.010	58	62
[26]	2015	2.91, 3.02	0.70 × 0.46 × 0.008	57.28	49
[28]	2016	2.4, 2.5	0.31 × 0.29 × 0.008	75	41.6
[27]	2016	6.37, 5.75	0.29 × 0.29 × 0.019	90	40
[29]	2016	3.38, 2.85	0.40 × 0.27 × 0.011	81	59.7
[31]	2017	3.25, 4.4	0.30 × 0.30 × 0.017	84	41
[35]	2017	1.85, 1.88	0.41 × 0.41 × 0.064	42.9	22.8
[32]	2018	6.03, 6.1	0.32 × 0.23 × 0.015	55	42
[34]	2017	2.86, 3.0	0.25 × 0.28 × 0.014	96.5	63.3
[33]	2017	3.12, 3.4	0.29 × 0.32 × 0.006	88	64.7
[36]	2018	6.42, 6.52	1.41 × 0.89 × 0.02	86.9	74.3
[37]	2018	3.48, 3.12	0.33 × 0.33 × 0.012	56	63.61
[38]	2018	2.82, 2.92	0.308 × 0.308 × 0.004	81	70.54
Proposed	2018	6.07, 3.60	0.29 × 0.29 × 0.012	126.85	73.33

**Table 2 sensors-18-04261-t002:** Scaling the dimensions with performance and applications.

Goal Frequency (GHz)	Theoretical Patch Dimension (mm)	Full Dimension (mm)	Relatives BW	AR Relatives BW	Applications
2.21	40.53 × 31.30 × 1.6	39 × 39 × 1.6	2.21–9.20 (122.80%)	2.28–4.72 (69.71%)	ISM, WiMAX, WLAN, satellite communications, and lower frequency bands CP type
1.15	77.90 × 60.57 × 1.6	78 × 78 × 1.6	1.15–2.58 (76.67%)	1.28–2.63 (65.52%)	GPS, WLAN, ISM Applications
1.48	60.53 × 46.98 × 1.6	58.7 × 58.7 × 1.6	1.48–6.37 (124.58%)	1.63–3.34 (68.81%)	GPS, GNS, ISM, WLAN Applications
2.92	30.68 × 23.55 × 1.6	29 × 29 × 1.6	2.92–10.60 (113.60%)	2.88–5.80 (67.28%)	S and C band Applications
4.84	23.55 × 13.62 × 1.6	19.5 × 19.5 × 1.6	4.84–12 (85%)	4.12–8.08 (64.94)	C band Applications

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
