# Peer review of "Circularly Polarized Broadband Printed Antenna for Wireless Applications"

_sensors, 2018, doi:10.3390/s18124261_

Reviewer 1 Report

The authors have presented wideband circualrly polarized printed antenna.

Results on the impedance and axial ratio bandwidths are good.

The radiation patterns are presented only for low frequencies (2.4 and 3.5 GHz).

The radiation patterns at higher frequencies should be presented (above 4 GHz).

Figure 4(d) should be improved.

Author Response

Dear Reviwer

Please find the enclosed revised manuscript. The revisions in the manuscript are marked in red color. In the revision, we have addressed the  reviewers’ comments. The replies to the comments are atatched here.

Thanks

Reviewer 2 Report

The proposed idea is interesting and the measurement results follow the simulated predictions. However, some aspects still need to be clarified before considering publication.

1. Some related researches are missing. Researches on UWB Microstrip antenna using RIS should be added: 

[1] L.-W. Li, Y.-N. Li, T. S. Yeo, J. R. Mosig, and O. J. Martin, “A broadband and high-gain metamaterial microstrip antenna,” Appl. Phys. Lett., vol. 96, no. 16, p. 164101, 2010.

[2] Y. Dong and T. Itoh, “Metamaterial-based antennas,” Proc. IEEE, vol.100, no. 7, pp. 2271-2285, Jul. 2012

[3] J. Yao, F. M. Tchafa, A. Jain, S. Tjuatja and H. Huang, “Far-field interrogation of microstrip patch antenna for temperature sensing without electronics,” IEEE Sensors J., vol. 16, no. 19, pp. 7053-7060. Oct. 2016

2. Could you please explain how does the slot circle on the ground plan help to improve the bandwidth of the proposed antenna?

3. There are several black blocks in fig.4, please modify. 

Author Response

Dear Reviewer

Please find the enclosed revised manuscript. The revisions in the manuscript are marked in red color. In the revision, we have addressed the  reviewers’ comments. The replies to the comments are atatched here.

Thanks

Reviewer 3 Report

This paper presented a design of wideband circularly polarized printed antenna. Basically, the presented antenna is absolutely similar to the design proposed in [22] with few minor changes, such as the dielectric substrate and the feedline shape, etc. Therefore, in the reviewer point of view, the paper does not show any novelty.

[22] Tuan Le, T.; Hoang The, V.; Chang Park, H. Simple and compact slot‐patch antenna with broadband circularly polarized radiation. Microwave Opt. Technol. Lett 2016, 58, 1634-1641.

Other comments:

1.      The authors mentioned that Ant1 is an LP antenna; however, as shown in Figures 3(a) and (b), Ant1 produced a CP operation at approximately 6.2 GHz. Please correct this.

2.      Please revise the Figure 4(d), the vectors are covered by the black box.

3.      Table 2 showed antenna performance at different frequency ranges by scaling the proposed antenna with the different values of scaling factor and the authors claimed that this may be the novelty of the presented antenna. However, this is very common technique in designing the antenna, nothing novelty here.

4.      The paper has numerous grammatical mistakes; hence a review of English grammar is necessary.

Author Response

Dear Reviewer

Please find the enclosed revised manuscript. The revisions in the manuscript are marked in red color. In the revision, we have addressed the  reviewers’ comments. The replies to the comments are attached here.

Thanks

Round  2

Reviewer 3 Report

The authors revised the paper well based on the reviewers' comments.